# Computer-Aided Design for Identifying Anticancer Targets in Genome-Scale Metabolic Models of Colon Cancer

**DOI:** 10.3390/biology10111115

**Published:** 2021-10-29

**Authors:** Chao-Ting Cheng, Tsun-Yu Wang, Pei-Rong Chen, Wu-Hsiung Wu, Jin-Mei Lai, Peter Mu-Hsin Chang, Yi-Ren Hong, Chi-Ying F. Huang, Feng-Sheng Wang

**Affiliations:** 1Department of Chemical Engineering, National Chung Cheng University, Chiayi 62102, Taiwan; easonchao1008@gmail.com (C.-T.C.); yu19941120@gmail.com (T.-Y.W.); a6901378tw@gmail.com (P.-R.C.); wwh@cs.ccu.edu.tw (W.-H.W.); 2Department of Life Science, Fu-Jen Catholic University, New Taipei City 24205, Taiwan; 062989@mail.fju.edu.tw; 3Department of Oncology, Taipei Veterans General Hospital, Taipei 11217, Taiwan; ptchang@vghtpe.gov.tw; 4Faculty of Medicine, National Yang Ming Chiao Tung University, Taipei 11211, Taiwan; 5Department of Biochemistry, Graduate Institute of Medicine, Kaohsiung Medical University, Kaohsiung City 80708, Taiwan; m835016@cc.kmu.edu.tw; 6Institute of Biopharmaceutical Sciences, National Yang Ming Chiao Tung University, Taipei 11211, Taiwan; cyhuang5@ym.edu.tw; 7Department of Biotechnology and Laboratory Science in Medicine, National Yang Ming Chiao Tung University, Taipei 11211, Taiwan

**Keywords:** metabolite-centric target, reaction-centric target, fuzzy optimization, two-target combination

## Abstract

**Simple Summary:**

Discovery of anticancer targets with minimal side effects is a major challenge in drug discovery and development. This study developed a fuzzy optimization framework for identifying anticancer targets. The framework was applied to identify not only gene regulator targets but also metabolite- and reaction-centric targets. The computational results show that the combination of a carbon metabolism target and any one-target gene that participates in the sphingolipid, glycerophospholipid, nucleotide, cholesterol biosynthesis, or pentose phosphate pathways is more effective for treatment than one-target inhibition is, and a two-target combination of 5-FU and folate supplement can improve cell viability, reduce metabolic deviation, and reduce side effects of normal cells.

**Abstract:**

The efficient discovery of anticancer targets with minimal side effects is a major challenge in drug discovery and development. Early prediction of side effects is key for reducing development costs, increasing drug efficacy, and increasing drug safety. This study developed a fuzzy optimization framework for Identifying AntiCancer Targets (IACT) using constraint-based models. Four objectives were established to evaluate the mortality of treated cancer cells and to minimize side effects causing toxicity-induced tumorigenesis on normal cells and smaller metabolic perturbations. Fuzzy set theory was applied to evaluate potential side effects and investigate the magnitude of metabolic deviations in perturbed cells compared with their normal counterparts. The framework was applied to identify not only gene regulator targets but also metabolite- and reaction-centric targets. A nested hybrid differential evolution algorithm with a hierarchical fitness function was applied to solve multilevel IACT problems. The results show that the combination of a carbon metabolism target and any one-target gene that participates in the sphingolipid, glycerophospholipid, nucleotide, cholesterol biosynthesis, or pentose phosphate pathways is more effective for treatment than one-target inhibition is. A clinical antimetabolite drug 5-fluorouracil (5-FU) has been used to inhibit synthesis of deoxythymidine-5${}^{\prime }$-triphosphate for treatment of colorectal cancer. The computational results reveal that a two-target combination of 5-FU and a folate supplement can improve cell viability, reduce metabolic deviation, and reduce side effects of normal cells.

## 1. Introduction

The process of drug discovery and development is challenging as well as cost and time intensive. Recent progress in omics fields (e.g., genomics [1], transcriptomics [2], proteomics [3], metabolomics [4], and fluxomics [5]) can promote the development of technology for discovering new drug targets [6]. Advancements in high-throughput data acquisition have been combined with systems biology approaches to increase the time and cost effectiveness of drug target discovery through computer-aided simulation techniques. Metabolism is the primary biological mechanism linking genotype with phenotype; studying metabolism facilitates understanding of cell physiology and disease phenotypes caused by metabolic dysregulation [7,8,9,10,11]. Genome-scale metabolic networks (GSMNs) relate metabolites and reactions and represent the full set of intracellular metabolic processes curated using knowledge of cellular functions from the literature. GSMNs combined with constraint-based approaches are leading approaches for developing methods to simulate cell behavior, such as flux balance analysis (FBA) [12,13].

Systems biology, a holistic approach to the study of biological systems, involves the modeling and analysis of metabolic pathways, regulatory networks, and signal transduction networks to understand cellular behavior. Cellular metabolism is often altered during disease; therefore, metabolic analysis can facilitate drug discovery. During tumor development, the metabolic processes of cancer cells are altered to sustain their uncontrolled proliferation. Due to progress in research in the last decade, metabolic reprogramming has become a common focus of cancer studies [14,15]. Numerous studies have employed metabolic rewiring to understand cancer-specific metabolic networks and to predict anticancer targets that could impair tumor growth or viability [9,10,16,17,18,19,20,21].

Several publications have applied cancer-specific GSMNs to identify anticancer targets [7,9,16,21,22,23,24,25,26,27,28,29]. The integration of omics data, cancer-specific GSMNs, and FBA has recently utilized the heterogeneity of metabolic patterns to discover biomarkers of cancers [30]. Mapping these tissue-specific metabolisms in GSMNs provides deeper insight into the metabolic basis of physiological and pathological processes. Current context-specific-model-building algorithms can be broadly categorized into flux-dependent methods and pruning methods [16,22,23,28,31,32,33,34,35,36,37]. Flux-dependent methods identify an optimal GSMN and include the maximum number of high-confidence reactions as supported by substantial experimental data. By contrast, pruning methods start with a core set of reactions obtained through literature reviews or experimental data and proceed by removing reactions from the reconstruction while maintaining functionality in a core reaction set.

Colorectal cancer (CRC) is a worldwide health burden and it is the third most common type of cancer and the fourth most common cause of cancer-related death [38]. An estimated 51,020 deaths from CRC were reported in 2019 in the United States [38]. In 2017, the Taiwan Cancer Registry reported that CRC was the most frequently diagnosed cancer in men and the second most frequently diagnosed cancer in women [39]. In our previous study, the human metabolic network Recon 2.2 [40] was incorporated with the Human Protein Atlas (HPA) [3] to reconstruct GSMNs for cancerous colorectal tissue and its healthy counterpart. An oncogene inference optimization algorithm was introduced to integrate both models to predict which dysregulated genes cause tumorigenesis [41]. The algorithm was also applied to identify oncogenes for head-and-neck cancer [42] and non-small-cell lung carcinoma [43]. This study introduced a tri-level optimization framework (TLOP) for identifying anticancer targets (IACT) for treatment of cancers. The IACT framework, extended from the oncogene inference optimization algorithm [41,42], identified not only gene regulator targets but also metabolite- and reaction-centric targets. Fuzzy set theory was also applied to investigate cell viability, metabolic deviation, and side effects after target treatment. CRC was used as a case study to illustrate the performance of IACT.

## 2. Materials and Methods

### 2.1. Reconstruction of Tissue-Specific GSMNs

This study combined a human metabolic network (Recon 3D) [12] with RNA-Seq expression data from The Cancer Genome Atlas (TCGA) database [44] to reconstruct population-based tissue-specific GSMNs for CRC and healthy counterpart tissue. The RNA-Seq data of 41 healthy colorectal samples with FPKM-UQ normalized expression value and 478 colon adenocarcinoma samples with different TCGA barcode were downloaded from the TCGA database and normalized using quantile normalization; then the mean, confidence interval, and coefficient of dispersion were calculated for each gene. This information was used to evaluate supportive genes and classify them into four groups (high, medium, low, and not detected). Based on the four gene groups and Recon 3D, the CORDA algorithm was used to reconstruct GSMNs for cancer and healthy tissue. We developed a systems biology program to automatically build stoichiometric and gene-protein-reaction models in the General Algebraic Modeling System (GAMS) to discover the anticancer targets with few side effects. The workflow of the reconstruction of the GSMM for CRC is shown in Figure 1.

### 2.2. Optimization Framework for Target Identification

We aimed to identify anticancer targets that not only are lethal to cancer cells but also minimize the side effects of toxicity-induced tumorigenesis for normal cells and have reduced metabolic perturbation. We established a TLOP to mimic a wet-lab experiment for identifying targets. The flowchart of the in silico experiment is displayed in Figure 2. The optimal design concept of the IACT framework is described in Table 1. The TLOP is a hierarchical optimization problem with four objectives and subject to inner optimization problems describing the characteristics of cancer cells for targeting treatment and metabolic perturbation of normal cells caused by treatment. The first objective is to evaluate the mortality of cancer cells, a common criterion for discovering target problems [18,26,45,46,47], that the biomass growth rate of cancer cells for treatment (denoted as treated cells) has a value as small as possible. Anticancer targets may cause toxicity-induced tumorigenesis in normal cells and lead to harmful metabolic perturbations (referred to as metabolic perturbation). Therefore, the metabolic perturbation is defined as the normal cells accompanied with treatment (referred to as perturbed cells) to alter their metabolic flux distributions. The second objective is to obtain superior cell viability of perturbed cells, that is equivalent to maximize the ATP production rate and minimize the cell growth rate. We defined two types of metabolic deviations for perturbed cells to evaluate a side-effect grade. They are the differences of flux distributions of perturbed cells from cancer template and normal template, respectively. The third objective is to keep the metabolic deviations of perturbed cells as dissimilar as possible to the cancer template, and the fourth objective is to keep the metabolic deviations of perturbed cells as similar as possible to the normal template. The four objectives are formulated based on the fuzzy set theory [48,49,50,51,52], and detailed in Appendix A. The constraint-based models for cancer and normal cells in inner optimization problems are expressed as follows: (1)$$\begin{array}{c} \mathrm{Inner}\;\mathrm{optimization}\;\mathrm{problems}:\\ \left\{\begin{array}{c} \mathrm{Treatment}\;\mathrm{of}\;\mathrm{cancer}\;\mathrm{cells}:\\ \begin{array}{c} \left\{\begin{array}{c} \mathrm{FBA}\;\mathrm{problem}\\ \max_{v_{f/b}}obj_{CA}\equiv \left(w_{ATP}v_{ATP}+w_{biomass}v_{biomass}\right)\\ \mathrm{subject}\;\mathrm{to}\\ \;N^{\mathrm{CA}}\left(v_{f}-v_{b}\right)=0\\ \;v_{f/b,i}^{LB,TR}\leq v_{f/b,i}\leq v_{f/b,i}^{UB,TR},z_{i}\in \Omega^{TR}\\ \;v_{f/b,j}^{LB}\leq v_{f/b,j}\leq v_{f/b,j}^{UB},z_{j}\notin \Omega^{TR} \end{array}\right. \end{array}\\ \begin{array}{c} \left\{\begin{array}{c} \mathrm{UFD}\;\mathrm{problem}\\ \min_{v_{f/b}}\sum_{i\in \Omega^{Int}}\left(v_{f,i}^{2}+v_{b,i}^{2}\right)\\ \mathrm{subject}\;\mathrm{to}\\ \;N^{\mathrm{CA}}\left(v_{f}-v_{b}\right)=0\\ \;v_{f/b,i}^{LB,TR}\leq v_{f/b,i}\leq v_{f/b,i}^{UB,TR},z_{i}\in \Omega^{TR}\\ \;v_{f/b,j}^{LB}\leq v_{f/b,j}\leq v_{f/b,j}^{UB},z_{j}\notin \Omega^{TR}\\ \;obj\geq obj_{CA}^{*} \end{array}\right. \end{array}\\ \mathrm{Perturbation}\;\mathrm{of}\;\mathrm{normal}\;\mathrm{cells}:\\ \begin{array}{c} \left\{\begin{array}{c} \mathrm{FBA}\;\mathrm{problem}\\ \max_{v_{f/b}}\mathrm{ob}j_{BL}\equiv \left(w_{ATP}v_{ATP}+w_{biomass}v_{biomass}\right)\\ \mathrm{subject}\;\mathrm{to}\\ \;N^{\mathrm{BL}}\left(v_{f}-v_{b}\right)=0\\ \;v_{f/b,i}^{LB,TR}\leq v_{f/b,i}\leq v_{f/b,i}^{UB,TR},z_{i}\in \Omega^{TR}\\ \;v_{f/b,j}^{LB}\leq v_{f/b,j}\leq v_{f/b,j}^{UB},z_{j}\notin \Omega^{TR} \end{array}\right. \end{array}\\ \begin{array}{c} \left\{\begin{array}{c} \mathrm{UFD}\;\mathrm{problem}\\ \min_{v_{f/b}}\sum_{i\in \Omega^{Int}}\left(v_{f,i}^{2}+v_{b,i}^{2}\right)\\ \mathrm{subject}\;\mathrm{to}\\ \;N^{\mathrm{BL}}\left(v_{f}-v_{b}\right)=0\\ \;v_{f/b,i}^{LB,TR}\leq v_{f/b,i}\leq v_{f/b,i}^{UB,TR},z_{i}\in \Omega^{TR}\\ \;v_{f/b,j}^{LB}\leq v_{f/b,j}\leq v_{f/b,j}^{UB},z_{j}\notin \Omega^{TR}\\ \;obj\geq obj_{BL}^{*} \end{array}\right. \end{array} \end{array}\right. \end{array}$$
where $v_{f/b}$ is the forward/backward flux vector of reversible reactions; the stoichiometric matrices $N^{\mathrm{CA}}$ and $N^{\mathrm{BL}}$ for tissue-specific cancer and normal cells, respectively, can be reconstructed by using Recon 3D [12] with the TCGA [44] or HPA [3] databases; $v_{f/b,i}^{LB}$ and $v_{f/b,i}^{UB}$ are the positive lower and upper bounds of the *i*th backward/forward flux, respectively; the integer vector z is used to determine mutated enzymes; and $obj_{CA/BL}^{*}$ is the maximum cellular objective obtained from FBA. The aim of the IACT framework is to determine modulated reactions for metabolite-centric and reaction-centric approaches as well as for the gene-centric approach. The approaches are dependent on the restrictions for the lower and upper bounds $v_{f/b,i}^{LB,TR}$ and $v_{f/b,i}^{UB,TR}$ of the *i*th modulated reactions in the inner optimization problem. The restrictions on the bounds are discussed in Appendix A.

### 2.3. Hierarchical Fitness in Outer Optimization

The IACT framework (Table 1) is expressed as a fuzzy multiobjective optimization problem (i.e., a hierarchical optimization problem). A bilevel optimization problem is a simple hierarchical optimization problem that converts the inner optimization problem into constraints in its outer-level problem by using duality theory. However, the inner problems in Equation (Equation 1) include two loops that are difficult to convert to constraints in the outer problem. The nested hybrid differential evolution (NHDE) algorithm was applied to predict oncogenes of various cancers [41,42]. This study extends the NHDE to solve the IACT problem. The computational procedures are presented in Appendix A and the implementation code in the GAMS (General Algebraic Modeling System) modeling language can be downloaded from https://chopin.ccu.edu.tw/?link=38a8SPuu6p (accessed on 1 July 2021). The outer optimization problem consists of three fuzzy goals and one crisp goal as shown in detail in Appendix A. In fuzzy set theory [52], fuzzy objectives can be attributed membership functions to convert the objectives into decision criteria and thus convert the fuzzy optimization problem into a maximizing decision-making problem [50].

A linear membership grade was applied to normalize each objective between zero and one; thus, the multiobjective functions were converted to a hierarchical fitness function for evaluating the fitness of the NHDE algorithm. The definition and computation for each membership grade are described in Appendix A, and the hierarchical fitness function is defined as follows:(2)$$\eta_{D}=\frac{\eta_{CV}+min\left(\eta_{CV},\eta_{DV}\right)}{2}$$
where the first priority grade $\eta_{CV}$ in the hierarchical fitness is the membership grade for the cell growth rates of the treated and perturbed cells. This grade is computed by the mean-min evaluation for both cells as follows:(3)$$\eta_{CV}=\frac{\frac{\eta_{CV}^{TR}+\eta_{CV}^{PB}}{2}+min\left(\eta_{CV}^{TR},\eta_{CV}^{PB}\right)}{2}$$
where the membership grade $\eta_{CV}^{TR}$ is the measure of the mortality of cancer cells in response to treatment, and the membership grade $\eta_{CV}^{PB}$ is an evaluation of cell viability for normal cells perturbed by treatment. This evaluation assesses the minimization of cell growth and maximization of ATP production for the perturbed cells and is computed by the mean-min evaluation of both grades as follows:(4)$$\eta_{CV}^{PB}=\frac{\frac{\eta_{biomass}^{PB}+\eta_{ATP}^{PB}}{2}+min\left(\eta_{biomass}^{PB},\eta_{ATP}^{PB}\right)}{2}$$
where $\eta_{biomass}^{PB}$ and $\eta_{ATP}^{PB}$ are the membership grades for the cell growth and ATP production of perturbed cells, respectively.

The second priority grade $\eta_{DV}$ in the hierarchical fitness function is used to evaluate metabolic deviations of the perturbed cells (the third and fourth goals of the framework) by using the following mean-min evaluation:(5)$$\eta_{DV}=\frac{\frac{\eta_{S_{F}}+\eta_{S_{M}}+\eta_{L_{F}}+\eta_{L_{M}}+\eta_{v}+\eta_{M}}{6}+min\left(\eta_{S_{F}},\eta_{S_{M}},\eta_{L_{F}},\eta_{L_{M}},\eta_{v},\eta_{M}\right)}{2}$$
where the membership grades $\eta_{S_{F}},\eta_{S_{M}},\eta_{L_{F}},$ and $\eta_{L_{M}}$ of the third goal are used to evaluate the maximization of the differences in the flux patterns of perturbed cells compared with templates generated from cancer and normal cells. The membership grades $\eta_{v}$ and $\eta_{M}$ of the fourth goal are used to measure the maximization of the flux and metabolite-flow similarities between perturbed cells and their normal counterparts. A higher membership grade for metabolic deviation implies a smaller metabolic perturbation due to treatment. The membership grade of side effects $\eta_{SE}$ is defined as follows:(6)$$\eta_{SE}=\frac{\frac{\eta_{CV}^{PB}+\eta_{DV}}{2}+min\left(\eta_{CV}^{PB},\eta_{DV}\right)}{2}$$
This membership grade was used to evaluate the metabolic deviation of the perturbed cells from templates generated from cancer and normal cells.

### 2.4. Factor Analysis

Log${}_{2}$ fold changes of all metabolite-flows for each perturbation (denoted as PB) to the normal (BL) state, and the template at cancer (CA) and normal states were described in Appendix A, and thus,
(7)$$\left\{\begin{array}{c} \mathrm{Perturbation}:L_{M,m}^{a}=\log_{2}\left(\frac{r_{m}^{PB_{a}}}{r_{m}^{BL}}\right),m\in \Omega^{m},a\in \Omega^{a}\\ \mathrm{Template}:L_{M,m}^{T}=\log_{2}\left(\frac{r_{m}^{CA}}{r_{m}^{BL}}\right) \end{array}\right.$$
where $\Omega^{m}$ is the set of metabolites in a GSMN, $\Omega^{a}$ is the set of identified anticancer targets, and the metabolite-flow rates $r_{m}^{PB/CA/BL}$ of the *m*th metabolite in each perturbation, cancer, and normal states were computed as follows:(8)$$r_{m}^{PB/CA/BL}=\sum_{i\in \Omega^{c}}\left(\sum_{N_{ij}>0,j}N_{ij}v_{f,j}^{PB/CA/BL}-\sum_{N_{ij}<0,j}N_{ij}v_{b,j}^{PB/CA/BL}\right),m\in \Omega^{m}$$
$\Omega^{c}$ is the set of metabolites located in different compartments of the cell. The expression enclosed in brackets in Equation (Equation 8) indicates the synthesis rate of the *i*th metabolite calculated by summing the influxes of the forward and backward reactions. Iterated principal factor analysis by using an orthogonal quartimax rotation method in SAS software (https://www.sas.com/, accessed on 1 July 2020) was used to analyze the $\log_{2}$ fold changes $\left(L_{M,m}^{T},L_{M,m}^{a};a\in \Omega^{a}\right)$ of the template and all perturbations.

## 3. Results and Discussion

### 3.1. Reconstruction of Healthy and Cancerous Models

The GSMN of Recon 3D consisted of 5835 species, 10,600 reactions, and 2248 associated genes. We retrieved RNA-Seq data of 41 healthy colorectal samples with FPKM-UQ normalized expression value and 478 colon adenocarcinoma samples with different TCGA barcode from the TCGA database to reconstruct tissue-specific GSMNs for healthy (HT) and cancer (CA) states. The CORDA algorithm was used to reconstruct the HT and CA colorectal models. The HT model comprised 3742 species, 6023 reactions, and 1934 genes, and the CA model comprised 4402 species, 7027 reactions, and 1920 genes. Both models were merged into the basal (BL) or normal model that was the union set of the HT and CA models and included 4541 species, 7453 reactions, and 1986 genes. The numbers listed in the overlapping regions of Figure 3 denote the number of identical species, reactions, and genes for the HT and CA models.

### 3.2. Gene-Centric Approach

The NHDE algorithm [41,42] was first applied to solve the IACT problem by using a gene-centric approach to identify optimal targets. The NHDE algorithm determined a set of one-target genes having the highest hierarchical fitness among 1934 candidates and the best 20 target genes, as shown in Table 2. The determined 17 out of 20 target genes are the same genes predicted by using the NCI-60 cancer cell lines [16]. This study identified three new targets genes *CRLS1*, *PGS1* and *ADSS2* for treatment. We downloaded the dataset of cancer cell lines from the Cancer Dependency Map (DepMap, https://depmap.org/portal/, accessed on 1 July 2020), and 51 colon cancer cell lines from the dataset (2021Q1 version) were collected. Surveying the dataset, we observed that most of the target genes could cause cell death for a high percentage of colon cancer cell lines except for *EBP*, *LSS*, and *NSDHL*. These genes participate in the cholesterol biosynthesis III pathway. Using the STRING database (https://string-db.org/, accessed on 1 July 2020) and a Markov clustering method, we classified these gene-encoded enzymes into four classes of protein-protein interaction (PPI) (Figure 4A) that participate in the sphingolipid, glycerophospholipid, nucleotide, cholesterol biosynthesis, and pentose phosphate pathways.

On the basis of this computation, we found that cancer cells died (cell growth rate $\leq 10^{-10}$) for each one-target gene treatment and that although the growth of normal cells was nearly zero, the ATP production rate was 63% of the maximum level. The cell viability grade $\eta_{CV}^{PB}$ for each perturbed cell was 0.625. Thus, the cell viability grade for treated and perturbed cells reached 0.719. However, the metabolic deviation grades $\eta_{DV}$ of these genes were greater than 0.65, except for that of *PGS1*, *PTDSS1*, and *DHODH*, which were less than 0.448. Therefore, side effect grades $\eta_{SE}$ for *PGS1*, *PTDSS1*, and *DHODH* were lower than those of other genes. A lower $\eta_{SE}$ implies more side effects, that is, the normal cells have a higher chance of tumorigenesis or significant metabolic perturbation due to the treatment.

The NHDE is a genetic algorithm that can obtain and rank targets with higher grade. We used two groups of candidates in the algorithm to identify multiple targets for reducing computational burden. The first candidate group includes 20 identified targets in Table 2, and the second group includes the other candidate genes. We performed a series of computations to obtain a set of two-target combinations and determine their optimal grades as shown in Figure 5. For each combination, cancer cells died and the ATP production rate of the perturbed cells was greater than 95%. Thus, cell viability grades were greater than 0.944 for all treatments except for combinations including *PCK1* (Figure 5). Most metabolic deviation grades $\eta_{DV}$ improved by approximately 5% compared with similar single-target treatments. All combinations with *PCK1* achieved cell viability grades of at least 0.67. These results demonstrate that cell viability grade can be reduced to improve metabolic deviation and reduce side effects. We used the identified enzyme-encoding genes to investigate the PPI network (Figure 4B) by using the STRING database. The PPI network has five classes of interaction that are similar to interactions of the one-target enzymes. We observed that treatment predictions were superior for two-target combinations that had one target involved in glycosaminoglycan metabolism or central carbon metabolism, and the other target involved in any of the cholesterol, sphingolipid, glycerophospholipid, or nucleotide pathways. The gene-encoded enzyme PTDSS1 combined with any one target in central carbon metabolism does not have better performance than one-target inhibitors did (Figure 5). We found the two-target combination of PTDSS1 and PTDSS2 increase metabolic deviation grade to 0.674 (or side effect grade of 0.676), but decrease cell viability grade to 0.762. We also identified a three-target combination (PTDSS1, PTDSS2, and ENO1) with cell variability grade of 0.953, metabolic deviation grade of 0.689, and side effect grade of 0.751. The side effect grade for the three-target combination has been a significant improvement, and the performance of treatment is nearly identical to those of two-targets shown in Figure 5.

The metabolic pathway modulated by the identified genes is shown in Figure 6. These identified genes could be modulated by many drugs from DrugBank [53]. Of the drugs retrieved from DrugBank, 26 act on DHODH and 20 act on HMG-CoA reductase (HMGCR) (Table 2). DHODH catalyzes the oxidation of dihydroorotate (dhor-S) to orotate (orot) by using ubiquinone as an electron acceptor. Orotate is then catalyzed by uridine monophosphate synthetase (UMPS) to generate uridine monophosphate (ump), which is essential for the de novo production of pyrimidines for RNA and DNA replication. UMPS was also identified by the IACT framework as shown in Table 2. Both DHODH and UMPS inhibition may be effective for treating CRC [54,55]. Some literature has suggested that DHODH can be used to treat other diseases such as small-cell lung cancer [56], acute myeloid leukemia [57], and autoimmune diseases [58]. On the basis of the computation, we observed that DHODH achieved a lower side effect grade ($\eta_{SE}=0.492$), implying that normal cells are more likely to undergo tumorigenesis or have significant metabolic perturbation due to the treatment.

DHODH was acted on by 26 drugs surveyed from DrugBank. Table 2 lists the number of available drugs for the identified one-target inhibitors. These drugs were used to investigate the grades of the adverse events by using a SIDER survey (http://sideeffects.embl.de/, accessed on 1 July 2020) and the ADDReSS (http://www.bio-add.org/ADReCS/, accessed on 1 July 2020) databases. The National Cancer Institute Common Terminology Criteria for Adverse Events (CTCAE) provide unique clinical descriptions of severity for adverse events (AE) from mild to death graded on a scale from 1 to 5. The average grade for each drug determined by using the CTCAE is shown in Appendix A. Of the 26 drugs acting on DHODH, 3 (leflunomide, atovaquone, and teriflunomide) have AE grades. The overall average AE grade was at most 10.04 (Table 2). This trend was consistent with the smaller computed $\eta_{SE}$ implying higher flux and metabolite-flow perturbations. Eight drugs (Appendix A) acting on HMGCR had an overall average AE grade of 8.77. The computed $\eta_{SE}$ was 0.637 for this target.

Two-target inhibitors, DHODH and PCK1, have been investigated as metabolic therapeutic targets for the treatment of CRC metastatic progression [55]. The IACT framework was applied to investigate the performance of the two-target enzymes in inhibiting treatments and to obtain a superior side effect grade ($\eta_{SE}=0.634$), as shown in Figure 5. This side effect grade improved by approximately 29% compared with that of one-target DHODH. However, the PCK1 treatment was unable to inhibit cancer cell proliferation according to the computations. PCK1 combined with the other genes such as UMPS was also evaluated and found to have better grades (Figure 5).

As discussed previously, PCK1 could improve reaction synergism and catalyze the reversible decarboxylation and phosphorylation of oxaloacetate (oaa) to produce phosphoenolpyruvate (pep), as shown in Figure 6. The metabolite pep is also produced by the conversion of 2-phosphoglycerate (2pg) catalyzed by α-enolase (ENO1). From Figure 5, we observed that two-target inhibition of DHODH and ENO1 had higher cell viability and metabolic deviation grades than the two-target inhibition of DHODH and PCK1 did. Combinations targeting ENO1 and the one-target genes in Table 2 also have higher grades. The identified target enzymes (e.g., GAPDH, PGK1, BPGM, and RPIA) involved in central carbon metabolism had similar results.

HMGCR, encoded by *HMGCR*, is a rate-controlling enzyme of the mevalonate pathway producing cholesterol and other isoprenoids (Figure 6). A total of 20 drugs targeting the enzyme HMGCR were retrieved from DrugBank. Cholesterol-lowering drugs targeting HMGCR are widely available and collectively known as statins. HMGCR has been investigated as an anticancer target for the clinical treatment of CRC, breast cancer, and other cancers [59,60,61,62]. Computations revealed that the one-target inhibitor could cause cancer cell toxicity and had a membership grade of 0.719 (Table 2). Two-target combinations targeting HMGCR and an enzyme participating in the central carbon metabolism (Figure 5) achieved higher cell viability grades ($\eta_{CV}>0.94$) than did HMGCR one-target treatments. Moreover, three enzymes (MVK, MVD, and PMVK) participating in the mevalonate pathway could block cancer cell growth, and treatments targeting these enzymes could achieve results approaching that of those targeting HMGCR.

### 3.3. Metabolite-Centric Approach

The IACT framework was also used for a metabolite-centric approach for identifying antimetabolites. One-target antimetabolites and two-target antimetabolites for treatment of CRC were determined as shown in Table 3 and Figure 7, respectively. HMGCR catalyzes the conversion of hydroxymethylglutaryl coenzyme A (hmgccoa) to mevalonate (mev-R)-a necessary step in the biosynthesis of cholesterol (Figure 6). A cell viability grade greater than 0.71 and metabolic deviation grade greater than 0.66 could be achieved with a side effect grade of 0.63 by blocking either metabolite (hmgcoa or mev-R) (Table 3).

The metabolites orot, dtmp, and ump are nucleotides participating in the pyrimidine biosynthetic pathway of DNA synthesis. If these metabolites were inhibited, grades of 0.7, 0.66, and 0.63, were achieved for cell viability, metabolic deviation, and side effects, respectively. The results were nearly identical to those obtained using the gene-centric approaches (Table 2 and Figure 6). We also determined other one-target metabolites achieving satisfactory grades for cellular viability and side effects (Table 3). These metabolites included glu-L and ser-L in amino acids, pe-hs in a class of glycerophospholipids, and crm-hs and sphmyln-hs in the ceramide and sphingolipid families, respectively. Serine is a precursor for numerous other metabolites, including sphmyln-hs and folate (fol), and is the principal donor of one-carbon fragments in biosynthesis. Serine is also important in metabolism in that it participates in the biosynthesis of purines and pyrimidines. In this study, these antimetabolites were identified by the IACT framework, as displayed with gray symbols in Figure 6.

Although DHODH inhibition could cause cancer cell toxicity, worse side effects were predicted. Targeting DHODH and either PCK1 or ENO1 could improve efficacy. DHODH inhibition blocks orotate (orot) production, whereas PCK1 and ENO1 produce pep. Two-target antimetabolite inhibition achieved grades of 0.936, 0.695, and 0.751 for cell viability, metabolic deviation, and side effects, respectively (Figure 7). These grades were nearly equal to those of the two-target enzyme inhibition for DHODH and ENO1 but superior than those for DHODH and PCK1. Statins, also known as HMGCR inhibitors, are drugs that block the conversion of mev-R to hmgcoa and thereby inhibiting cholesterol synthesis. We found that the two-target combinations of hmgcoa or mev-R with a metabolite in the central carbon metabolism such as g3p, 3pg, 2pg, pep, or pyr could be applied to treat CRC satisfactorily (Figure 7). From the data displayed in Figure 7, we identified various two-target antimetabolite combinations and observed that the highest grades were for treatments targeting glu-L with central carbon metabolism metabolites such as g3p, 3pg, 2pg, pep, and pyr.

To verify the performance of the IACT framework, we evaluated cell viability and side effect grades of perturbation for normal cells treated with a clinical anticancer drug, 5-fluorouracil (5-FU). 5-FU is an antimetabolite drug commonly used for the treatment of CRC [63]. The drug inhibits the biosynthesis of dtmp (Figure 6) by acting on a key enzyme, thymidylate synthetase (TYMS). We used IACT for dtmp synthesis inhibition and computed a cell viability grade of 0.719 and a metabolic deviation grade of 0.665. We also evaluated two-target metabolite combination of dtmp and fol and found that inhibiting dtmp and enhancing fol synthesis achieved a cell viability grade of 0.978 but reduced the metabolic deviation grade to 0.597 (Figure 7). However, the side effect grade increased to 0.678. The reduction of $\eta_{DV}$ was consistent with observations in review articles [63,64]. To increase the anticancer activity of 5-FU, a modulation cotreatment with leucovorin to enhance intracellular levels of folate has been used [64,65,66]. This strategy has been demonstrated to increase the in vitro and in vivo toxicity of 5-FU for numerous cancer cell lines [67,68,69].

Although folate is a substrate for DNA synthesis in cell cycle S phase, the combination of folate and 5-FU results in synthetic anti-cancer effects due to the shift in the dump/dtmp metabolism and sensitization of 5-FU cytotoxicity. Such metabolism reprogramming can be predicted from our model. In clinical trial for CRC patients, similar results can also be observed especially in 5-FU continuous infusion which enhancing thymidylate synthase inhibition. The combination regimen of 5-FU/folinic acid increased anti-tumor effects without increasing toxicities while 5-FU was continuously infused. In EORTC40952 clinical trial for untreated metastatic CRC patients, Köhne et al. [70] showed significantly longer progression free survival in 5-FU continuous infusion plus leucovorin group (5.6 months vs. continuous 5-FU alone 4.1 months vs. bolus 5-FU plus leucovorin 4.0 months, p=0.029) and similar stomatitis was observed between continuous 5-FU plus leucovorin (5%) and continuous 5-FU alone (3%), while especially high in 5-FU bolus plus leucovorin group (11%).

### 3.4. Factor Loading of Identified Targets

The $\log_{2}$ fold changes of metabolite-flows were used to form a 1478×201 matrix that excluded unchanged flows and included the identified anticancer target genes, identified antimetabolites, and the template. The matrix data were analyzed through factor analysis to obtain nine factors and their associated factor loadings as listed in Appendix A. The key concept of factor analysis is that multiple observed variables have similar response patterns because they are all associated with a latent (i.e., not directly measured) variable. According to the analysis, the first and second factor loadings (Figure 8) were 64.19% and 19.32%, respectively. Therefore, these two factors could explain the key flux alterations for perturbations. The first group around Factor 1 displayed in the red circle of Figure 8 contained 117 anticancer target genes and antimetabolites from the two-target combinations and three-target combination of (PTDSS1, PTDSS2, ENO1). The flux alternations for genes in the first group (Appendix A) are very similar, because the values of cell viability grades, metabolic deviation grades, and side effect grades (Figure 5 and Figure 7) of all genes in the group are close to each other. The mean fold changes of metabolite-flows for the first group are displayed in the green boxes of Figure 6A. We observed that the metabolite-flows in the central carbon pathway were reduced in comparison with others. By contrast, the template increased as indicated in the yellow boxes of Figure 6A. The key metabolic reprogramming in cancer cells is rapid glucose metabolism to pyruvate, which is then largely converted to lactate. Glutamine is then replenished by the TCA cycle for proliferation requirement [14,15]. From Figure 6A, we observed that the flux alterations of the template coincide with the metabolic reprogramming of cancer progression. Moreover, the fold changes of metabolite-flows for the perturbed cells were different from the template.

The second group around Factor 2 contained 42 anticancer target genes and antimetabolites. Half of these were from two-target combinations. These two-target combinations involved the enzyme PCK1. The mean fold changes of metabolite-flow for the second group (Figure 6B) are little different from the results for the first group. Furthermore, the factor loadings could explain relationship with the membership grades for the identified targets. The factor loadings for PTDSS1, (PTDSS1, PTDSS2), and (PTDSS1, PTDSS2, ENO1) are shown by red dots in Figure 8. The identified targets have different cell viability, metabolic deviation, and side effect grades. The grades of the three-target combinations were very close to those of the others in the first group. Moreover, the grades for HMGCR (Table 2) are different to those (Figure 5) of the two-target combination (HMGCR, ENO1), so that they are separated into two groups in Figure 8. This result indicates that the $\log_{2}$ fold changes of metabolite-flows for HMGCR can be discriminated from those of (HMGCR, ENO1).

## 4. Conclusions

A TLOP was designed to identify anticancer targets for the treatment of colon cancer. This IACT framework was applied to not only determine gene regulator drug targets but also discover metabolite- and reaction-centric targets. For the gene-centric approach, we determined one-target-gene-encoding enzymes that participate in the sphingolipid, glycerophospholipid, nucleotide, cholesterol biosynthesis, and pentose phosphate pathways. For two-target treatments, combinations of any of the one-target inhibitors and an enzyme (e.g., GAPDH, PGK1, ENO1, RPIA, BPGM, or PCK1) in the central carbon metabolism achieved better performance than one-target inhibitors did. For the metabolite-centric approach, 10 one-target metabolites (two fatty acids, three nucleotides, two amino acids, two sphingolipids, and one glycerophospholipid) were determined. These targets had nearly identical cell viability and side effects compared with those of one-target enzymes. To examine the performance of the IACT framework, cellular viability and side effects due to metabolic perturbation of normal cells after treatment with a clinical antimetabolite drug (5-FU) were investigated using the model. The computational results were consistent with 5-FU inhibiting dtmp synthesis to block cancer cell proliferation and with folate supplementation improving cell viability, slightly reducing metabolic deviation, and reducing side effects in comparison with the one-target 5-FU treatment [64,65,66]. These computational results are supported by in vitro experimental observations.

## Figures and Tables

**Figure 1 biology-10-01115-f001:**
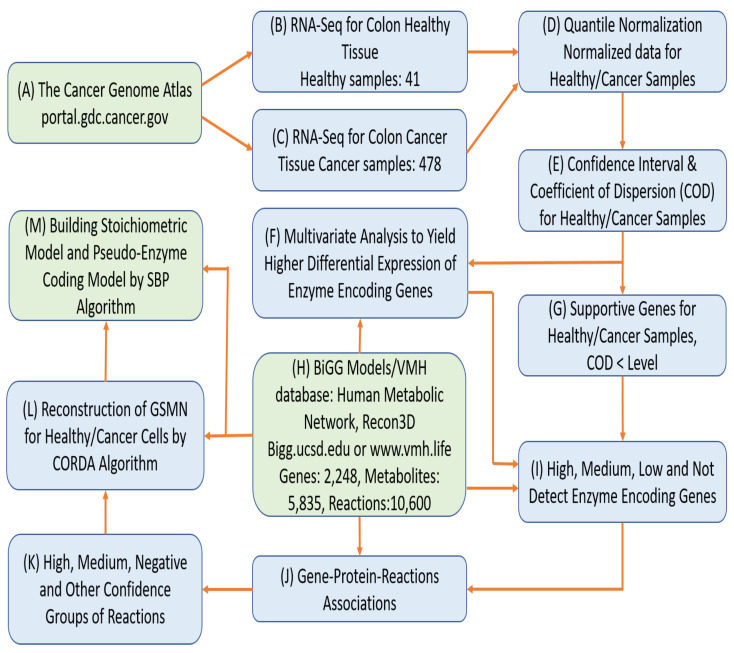
Roadmap of the reconstruction of genome-scale metabolic models. Workflows for reconstructing genome-scale metabolic models for CRC and its corresponding healthy tissue. (A) Download RNA-Seq data of CRC from the TCGA database. (B–G) A series statistical analyses of the download RNA-Seq data. (H) Download general human metabolic network model (Recon3D) from Bigg Models or VMH database. (I) Integrate the Recon3D model with the RNA-Seq data to classify all enzyme-encoded genes into four classes. (J) Retrieve gene-protein-reaction associations from the Recon3D model. (K) Compute the confidence score for each reaction based on the gene-protein-reaction associations, and classify all reactions into four groups. (L) Reconstruct tissue-specific metabolic models for healthy and cancer cells using the CORDA algorithm. (M) Create GAMS codes for tissue-specific metabolic models using the SBP platform.

**Figure 2 biology-10-01115-f002:**
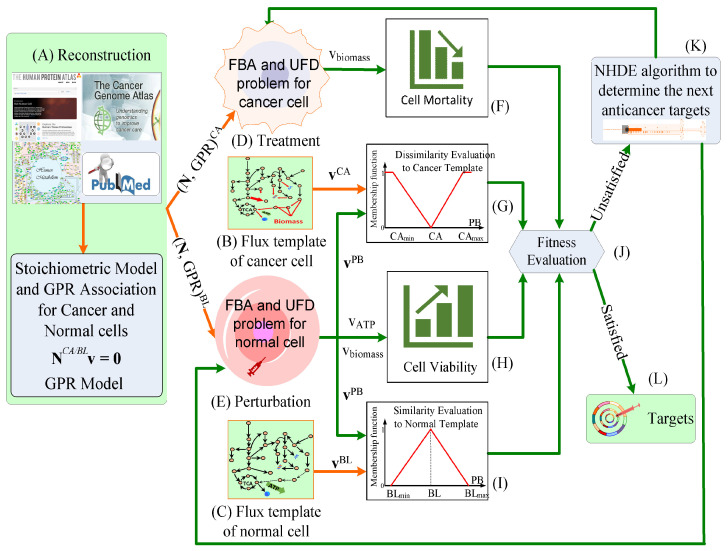
Work flowchart for identifying anticancer target framework. (**A**) Tissue-specific genome-scale metabolic models of cancerous (CA) and normal (BL) cells were reconstructed through biological data. (**B**) Flux distribution patterns for cancer tissue can be provided from clinical data if available; otherwise the template can be computed through FBA and UFD problem without considered dysregulated restriction. (**C**) Flux distribution patterns for normal tissue can be provided from clinical data if available; otherwise the template can be computed through FBA and UFD problem without considered dysregulated restriction. (**D**) A set of anticancer targets are identified by the nest hybrid differential algorithm (NHDE), and provided to compute the flux distributions for each cancer treatment. (**E**) The same targets are provided to compute the perturbated flux distributions of normal cell during treatment. (**F**) Using cancer cell growth rate, cell mortality is evaluated. (**G**) Using membership function, cancer template and perturbated fluxes are used to compute dissimilarity grade. (**H**) Cell viability of perturbed cell is computed using ATP synthesis and cell growth rate. (**I**) Using membership function, normal template and perturbated fluxes are used to compute similarity grade. (**J**) The four-objective grades are used to evaluate fitness for making decision in the NHDE algorithm. (**K**) The next anticancer targets are generated in the NHDE algorithm if the fitness is unsatifactory, and repeat the procedures. (**L**) The optimal targets are obtained if the fitness is satisfactory.

**Figure 3 biology-10-01115-f003:**
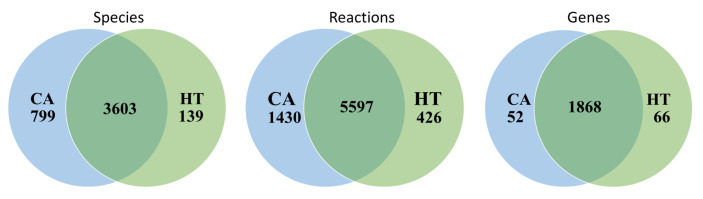
Comparison of metabolic network data between HT and CA models. Statistics of cancer (CA) and healthy (HT) reconstructed metabolic models. The basal (BL) model is the union set of the CA and HT models.

**Figure 4 biology-10-01115-f004:**
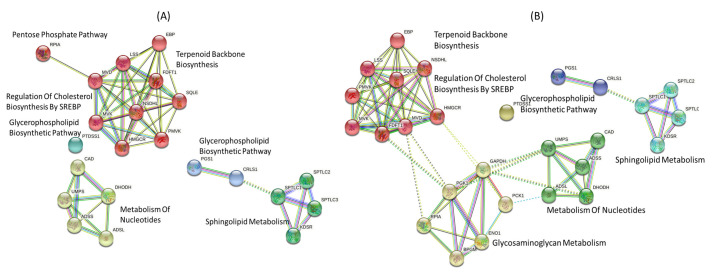
Protein-protein interactions. Protein-protein interactions of identified (**A**) one-target anticancer genes and (**B**) two-target combinations. MCL clustering in the STRING database was applied to classify one-target enzymes into four classes and two-target enzymes to five classes. The first class contained nine enzymes in terpenoid backbone biosynthesis, the second class included five enzymes in metabolism of nucleotides, the third class had four enzymes in sphingolipid metabolism, and the fourth class included two enzymes in glycerophospholipid biosynthetic pathway. For the one-target case, RPIA in the pentose phosphate pathway was categorized in the first class, but that in the two-target case was considered to participate in glycosaminoglycan metabolism or central carbon metabolism.

**Figure 5 biology-10-01115-f005:**
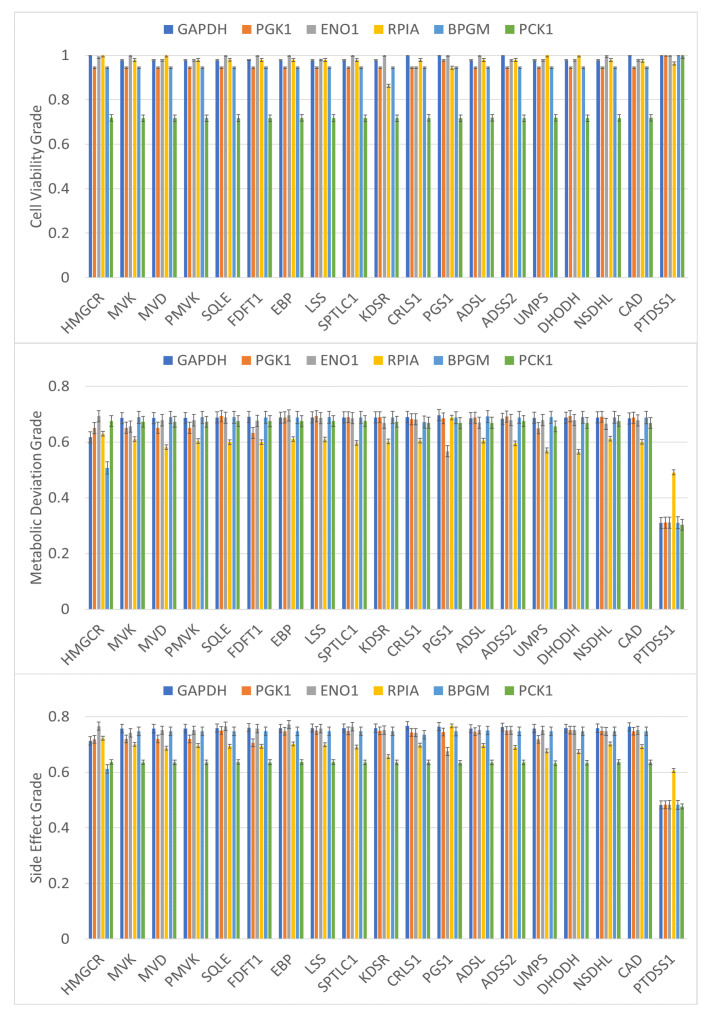
Membership grades for two-target combinations. Membership grades of cell viability ($\eta_{CV}$), metabolic deviation ($\eta_{DV}$), and side effect ($\eta_{SE}$) for two-target combinations of anticancer enzymes for colon cancer treatment. Cancer cell cytotoxicity was observed for each treatment. Therefore, the cell viability grade represents the cell growth viability of normal cells during treatment. Higher metabolic deviation grades indicate that the flux pattern of the perturbed cells was more dissimilar to the cancer template and more similar to the normal template. Higher side effect grades indicate fewer side effects. The numbers of drugs identified from DrugBank acting on each first target are shown in Table 2. The numbers of drugs acting on second targets are listed in brackets as follows: GAPDH (7), PGK1 (6), ENO1 (6), RPIA (1), BPGM (Not Available), and PCK1 (6). Error bar around each estimate was obtained through ten repeated executions.

**Figure 6 biology-10-01115-f006:**
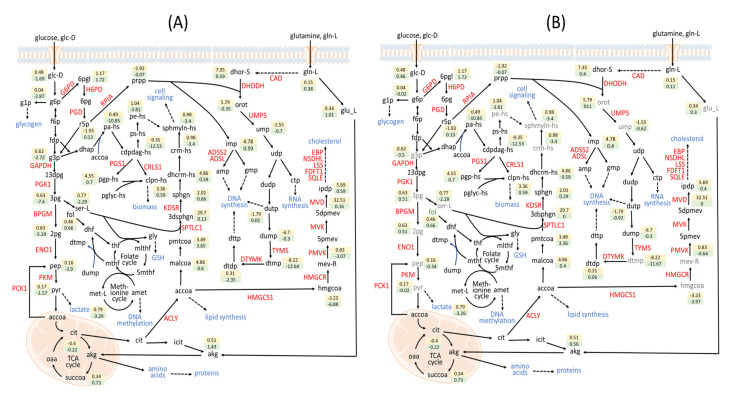
Mean fold changes of metabolite-flows for perturbed cases and the template. The $\log_{2}$ fold changes of metabolite-flows for the identified targets in (**A**) the first group and (**B**) the second group. The identified gene-encoding enzymes are represented in red. The identified antimetabolites are presented in gray. The values in yellow boxes denote $\log_{2}$ fold changes of metabolite-flows $\log_{2}\left(r_{m}^{CA}/r_{m}^{BL}\right)$ for the template, whereas the green box displays $\log_{2}\left(r_{m}^{MU_{a}}/r_{m}^{BL}\right)$ for each perturbation. A positive value indicates that the metabolite-flow of cancer cells or the perturbation is higher than basal. A negative value represents lower flow.

**Figure 7 biology-10-01115-f007:**
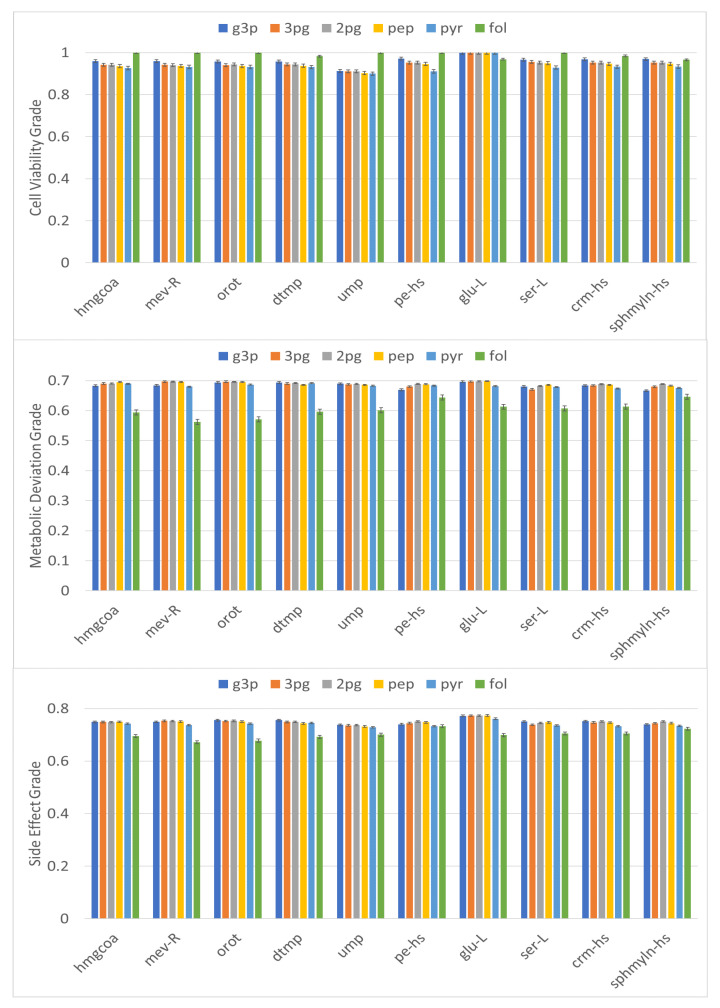
Membership grades for two-target combinations of antimetabolites. Membership grades of cell viability ($\eta_{CV}$), metabolic deviation ($\eta_{DV}$), and side effect ($\eta_{SE}$) for two-target combinations of antimetabolites for colon cancer treatment. Cancer cell cytotoxicity was observed for each treatment. Inhibition of two-target antimetabolites (except folate enhancement) improved membership grades compared with one-target counterparts. Glutamate combined with the metabolites in the central carbon metabolism such as g3p, 3pg, 2pg, pep, and pyr had the highest grades. Error bar around each estimate was obtained through ten repeated executions.

**Figure 8 biology-10-01115-f008:**
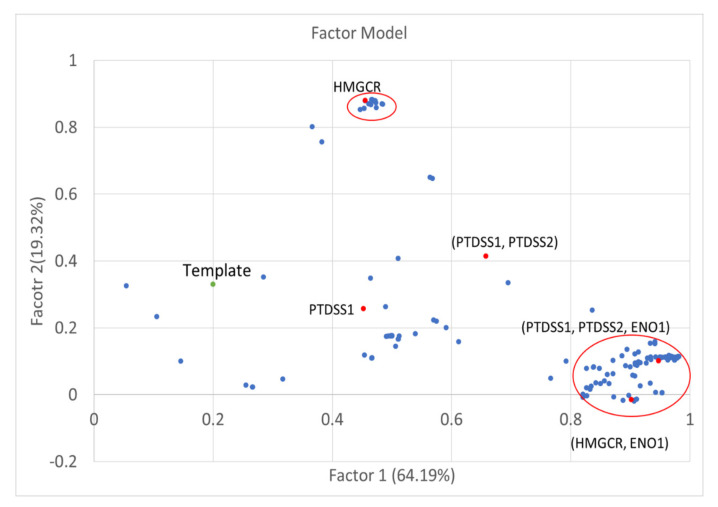
Membership grades for two-target combinations. First and second factor loadings of the identified anticancer target genes and antimetabolites. The template is the green dot. The first group around Factor 1 in the red circle contained 116 two-target combinations and a three combination of (PTDSS1, PTDSS2, ENO1) comprising anticancer target genes and antimetabolites. The second group around Factor 2 contained 42 anticancer target genes and antimetabolites. Half of these were two-target combinations. These two-target combinations included the enzyme PCK1.

**Table 1 biology-10-01115-t001:** Optimization framework for IACT by evaluating the performance of identified targets according to four objectives.

**Objectives in the outer optimization problem**
1. The first objective is to measure the mortality of treated cancer cells.
2. The second objective is to obtain superior cell viability of normal cells in cancer treatment.
3. The third objective is to keep the metabolic deviation of perturbed cells as dissimilar as possible to the cancer template.
4. The fourth objective is to keep the metabolic deviation of perturbed cells as similar as possible to the normal template.
**subject to the constraint-based models in the inner optimization problems**
1. FBA and UFD problems for treated cancer cells.
2. FBA and UFD problems for perturbation of normal cells due to cancer treatment.

**Table 2 biology-10-01115-t002:** Top 20 one-target genes obtained using the IACT framework. *SPTLC1/2/3* is a complex of serine palmitoyltransferase constructed by *SPTLC1*, *SPTLC2*, and *SPTLC3*. Other genes each encode for a single enzyme, as shown in the abbreviations section. The symbol (–) denotes that data are unavailable.

Gene	$\eta_{\mathrm{CV}}{\;}^{†}$	$\eta_{\mathrm{DV}}{\;}^{‡}$	$\eta_{\mathrm{SE}}{\;}^{§}$	N/D ${}^{♭}$	nDrugs ${}^{♮}$	Ave. AE ${}^{♯}$	Pathway ${}^{¶}$
*HMGCR*	0.719	0.675	0.637	50/51	20	8.77	Cholesterol Biosynthesis, Statin Pathway, Mevalonate Pathway
*MVK*	0.719	0.669	0.636	50/51	1	–	Mevalonate Pathway, Regulation of Cholesterol Biosynthesis By SREBP
*MVD*	0.719	0.669	0.636	48/51	–	–	Mevalonate Pathway, Regulation of Cholesterol Biosynthesis By SREBP
*PMVK*	0.719	0.669	0.636	23/51	–	–	Mevalonate Pathway, Regulation of Cholesterol Biosynthesis By SREBP
*SQLE*	0.719	0.672	0.637	3/51	4	6.73	Cholesterol Biosynthesis III, Statin Pathway
*FDFT1*	0.719	0.674	0.637	8/51	1	–	Cholesterol Biosynthesis III, Statin Pathway
*EBP*	0.719	0.674	0.637	0/51	1	8.94	Cholesterol Biosynthesis III
*LSS*	0.719	0.672	0.637	0/51	2	–	Cholesterol Biosynthesis III
*NSDHL*	0.719	0.674	0.637	0/51	1	–	Cholesterol Biosynthesis III
*SPTLC1/2/3*	0.719	0.670	0.636	37/51	2	–	Sphingolipid Metabolism
*KDSR*	0.719	0.670	0.636	6/51	–	–	Sphingolipid Metabolism
*CRLS1*	0.719	0.669	0.636	38/51	–	–	Glycerophospholipid Biosynthesis
*PGS1*	0.719	0.448	0.492	50/51	–	–	Glycerophospholipid Biosynthesis
*PTDSS1*	1.0	0.303	0.477	8/51	1	–	Glycerophospholipid Biosynthesis
*ADSL*	0.719	0.669	0.636	49/51	–	–	Metabolism of Nucleotides, Purine Metabolism
*ADSS2*	0.719	0.669	0.636	37/51	3	–	Metabolism of Nucleotides, Purine Metabolism
*UMPS*	0.719	0.655	0.632	29/51	2	9.82	Metabolism of Nucleotides, Pyrimidine Biosynthesis
*DHODH*	0.719	0.448	0.492	19/51	26	10.04	Metabolism of Nucleotides, Pyrimidine Biosynthesis
*CAD*	0.719	0.669	0.636	27/51	3	8.9	Metabolism of Nucleotides, Pyrimidine Biosynthesis
*RPIA*	0.719	0.675	0.637	6/51	1	–	Pentose Phosphate Pathway

${}^{†}$ Cell viability grade as evaluated from cancer cell treatment and the perturbations of normal cells due to treatment; ‡ Metabolic deviation grade indicating the perturbance of the cellular flux patterns as measured by dissimilarity to the cancer template and similarity to the basal template; ${}^{§}$ Side effect grade. A higher $\eta_{SE}$ indicates fewer predicted side effects; ${}^{♭}$ The cell death number (N) divided by the total number of colon cancer cells (D) used for the test from DepMap; ${}^{♮}$ The number of drugs retrieved from DrugBank that modulate each gene; ${}^{♯}$ Average grade of adverse events for drugs acting on an identified gene; ${}^{¶}$ Accessed from GeneCards.

**Table 3 biology-10-01115-t003:** Membership grades of cell viability ($\eta_{CV}$), metabolic deviation ($\eta_{DV}$), and side effect ($\eta_{SE}$) for the top 10 one-target antimetabolites determined by the IACT framework.

Metabolite	Symbol	$\eta_{\mathrm{CV}}$	$\eta_{\mathrm{DV}}$	$\eta_{\mathrm{SE}}$	Subclass
Hydroxymethylglutaryl Coenzyme A	hmgcoa	0.712	0.668	0.636	Fatty acyl thioesters
Mevalonate	mev-R	0.719	0.674	0.637	Fatty acids and conjugates
Orotate	orotsumpplementation	0.718	0.673	0.637	Pyrimidines and pyrimidine derivatives
Deoxythymidine-5${}^{\prime }$-Phosphate	dtmp	0.719	0.665	0.635	Pyrimidine deoxyribonucleotides
Uridine-5${}^{\prime }$-Monophosphate	ump	0.701	0.67	0.626	Pyrimidine ribonucleoside monophosphates
Phosphatidylethanolamine	pe-hs	0.795	0.691	0.7	Glycerophospholipids
L-Glutamate	glu-L	0.783	0.663	0.675	Amino acid
L-Serine	ser-L	0.789	0.656	0.672	Amino acid
N-Acylsphingosine	crm-hs	0.75	0.664	0.665	The parent compounds of the ceramide family
Sphingomyelin	sphmyln-hs	0.75	0.655	0.658	Phosphosphingolipids, fatty acyl group

## Data Availability

All data generated or analyzed during this study are included in this published article.

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
