# Peer review of "Computer-Aided Design for Identifying Anticancer Targets in Genome-Scale Metabolic Models of Colon Cancer"

_biology, 2021, doi:10.3390/biology10111115_

Round 1

Reviewer 1 Report

This manuscript revolves around 5-fluorouracil (5-FU) an "anti-folate!" and predicted reversal of toxicity by folate while 5-FU is an antimetabolite of folate metabolism

First of all I am extremely impressed by the methodology of this very systematic study. The inferences drawn are absolutely correct unless you know cancer biology or mechanism of action of 5-FU.

Folate and 5-fluorourasil are antagonists.. the authors bundled 4-5 key papers in the discussion using them to support their claim. If they had collaborated with an expert of biological aspects or went through the papers they casually cited they would know the relation between the two.

While there is no surprise folate reverses 5-FU toxicity. Not only that it reverses it's effect too!

Even with this grand blunder I am recommending this manuscript for major revision and not rejecting because the results are novel and need better interpretations.

There are two unanswered questions in the field:

  1. How folate an antagonist of 5-FU increases its toxicity as well as anti-neoplastic activity during cancer treatment.?
  2. What is the mechanism of folate that both collapses as well as promotes cancer (It's rather famous paradox).

Therefore if authors can answer these two questions from their rather expansive analyses and describe relation between 5-FU and Folate based on human trials and latest ex-vivo papers in introduction itself.

In discussion the results need to be better anchored on the existing data especially clinical trials.

Reviewer 2 Report

Figure 4 and Figure 6 should have error bars added

Reviewer 3 Report

In this manuscript, the authors aimed to

-minimize side effects causing toxicity-induced tumorigenesis on normal cells with smaller metabolic perturbations.

It is an interesting study, however something that I missed, why the authors tried to tackle this issue with smaller metabolic perturbations? What is the advantage of smaller metabolic perturbations during cancer treatment? Is there a correlation between toxicity and high metabolic perturbation? Please also elaborate on the metabolic perturbation, definition, and measurement. How did the authors calculate the metabolic perturbation?

It would be nice if the authors mention the metabolic heterogeneity in cancer metabolism which leads to personalize treatment. (https://www.mdpi.com/2075-4426/11/6/496)

I recommend the authors move Figure S1 to the main text. It would help the readers to better follow the method.

The number of genes, metabolites, and reactions of Recon3D that the authors used are different from the original model on the website. The number of genes and reactions is less than the numbers on the website and the number of metabolites is more. How the authors trimmed the original model? And why is the number of metabolites is more?

Original Recon3l from vmh

Recon3D in the paper

No. of Genes

3695

2247

No. of Reactions

13543

5835

No of Metabolites

4138

10600

How did the authors validate the reconstructed models by CODRA? Did they also constrain the exometabolites of models?

What the UFD stands for?

The authors calculated the metabolic pattern of cancer and normal tissues by FBA. Why not FVA? There are big issues with this approach. FBA returns just a random value taken from the distribution of allowed fluxes. That means that using a single vector of fluxes as returned by the solver is not meaningful without knowing the distribution of possible values. In order to use ratios as a meaningful score, authors should compute the distributions of valid fluxes (FVA) and compare if there are statistically significant changes between models. There are not only many valid fluxes for a given network (see for example https://www.sciencedirect.com/science/article/abs/pii/S1096717603000582), but also many different possible optimal networks for a given reconstruction (for example https://journals.plos.org/ploscompbiol/article?id=10.1371/journal.pcbi.1005568).

How did the authors evaluate cell mortality by using cancer cell growth rate?

Round 2

Reviewer 1 Report

There are papers reporting increased folate resulting in 5FU ineffectiveness and increased toxicity. It's rather interesting to see the same at a molecular level from this data. 

Reviewer 3 Report

The authors addressed all my concerns and I would be happy to recommend this manuscript for publication.